# SRMRI: A Diffusion-Based Super-resolution Framework and Open Dataset for Blind MRI Super-Resolution

**Arpan Poudel**[1]                                        ARPANP@UARK.EDU
**Mamata Shrestha**[1]                                    MAMATAS@UARK.EDU
**Nian Wang**[2]                              NIAN.WANG@UTSOUTHWESTERN.EDU
**Ukash Nakarmi**[1]                                    UNAKARMI@UARK.EDU

[1] *University of Arkansas, Fayetteville, Arkansas, USA*

[2] *University of Texas Southwestern Medical Center, Dallas, Texas, USA*

**Editors:** Accepted for publication at MIDL 2025

## Abstract

Existing deep learning methods for medical image super-resolution (SR) often rely on paired datasets generated by simulating low-resolution (LR) images from corresponding high-resolution (HR) scans, which can introduce biases and degrade real-world performance. To overcome these limitations, we present an unsupervised approach based on a score-based diffusion model that does not require paired training data. We train a score-based diffusion model using denoising score matching on HR Magnetic Resonance Imaging (MRI) scans, then perform iterative refinement with a stochastic differential equation (SDE) solver while enforcing data consistency from LR scans. Our method provides faster sampling compared to existing generative approaches and achieves competitive results on key metrics, though it does not surpass fully supervised baselines in PSNR and SSIM. Notably, while supervised models often report higher numerical metrics, we observe that they can produce suboptimal reconstructions due to their reliance on fixed upscaling kernels. Finally, we introduce the SRMRI dataset, containing LR and HR images obtained from scanner for training and evaluating MR image super-resolution models. Code and dataset are available at: https://github.com/arpanpoudel/SRMRI.

**Keywords:** unsupervised MRI super-resolution, MRI reconstruction, super-resolution dataset, score-based diffusion model

## 1. Introduction

MRI is a widely used technology in medical imaging that acquires data in k-space—the Fourier domain—by placing the subject on a magnetic field and generating corresponding signals. Images are then obtained using inverse Fourier transform on k-space data. Although HR images are desired for accurate clinical diagnosis, their acquisition is limited by hardware limitations, patient movement, and extended scan times, leading to a reliance on LR images. Super-resolution techniques provide a reliable solution to overcome these challenges.

Supervised deep learning method for medical image super-resolution (de Leeuw den Bouter et al., 2022; Isaac and Kulkarni, 2015; Sano et al., 2017) directly learns the mapping between LR and HR images by training on a large paired dataset. These paired datasets are prepared by simulating specific degradation methods such as Bicubic Downsample, Gaussian Blurring, or Median Filtering (Keys, 1981) on HR images to obtain LR images.

In MRI super-resolution, acquiring matching LR and HR image pairs is often impossible due to practical challenges. These include subject motion between acquisitions and excessively long scanning times, making it difficult to obtain exact training pairs. Additionally, the degradation process in MRI is non-deterministic making simulated data unsuitable for practical cases. In real-world scenarios, high-resolution images may degrade due to several factors such as magnetic field inhomogeneity, improper acquisition parameters, patient motion during the scan, and scanner electrical noise or insufficient acquisition time leading to low-resolution images that cannot be accurately mimicked by simulated LR images. Consequently, deep learning models trained on specific degradation methods are biased (Shrestha et al., 2023) on the training data pair and only learn to super-resolve the training data degradation. When the simulated degradation method or degradation factor changes, we have to re-obtain the paired dataset and re-train the model.

In this work, we introduce the SRMRI dataset, a novel collection of LR and HR MRI images obtained directly from the scanner. Further, we also propose an unsupervised framework for reconstructing HR MRI images that does not require a paired training dataset for training, thus avoiding the limitation of a fixed degradation process. Our method is inspired by the idea of learning the prior distribution of HR images by training a score-based diffusion model (Song et al., 2020) as the data prior and provide a sampling algorithm to sample images from data distribution that are consistent with the low-resolution images.

Our work makes the following contributions:

- We introduce the SRMRI dataset, a collection of low-resolution and high-resolution images on the same subjects acquired from the scanner.

- We train a score-based diffusion model on MRI images to generate HR MRI images as unconditional samples using a numerical solver. We propose a sampling algorithm to reconstruct HR images from the LR images by alternating between Diffusion Posterior Sampling (DPS) (Chung et al., 2023) and image fusion strategy. Finally, we evaluate our model on scanner-obtained LR-HR pairs to provide a better representation of performance.

An overview of our method is illustrated in Figure 1, and the detail of our method in Section 3.

## 2. Related Works

### 2.1. Score-based Diffusion Model

We can construct a diffusion process $\{\boldsymbol{x}(t)\}_{t=0}^1$ on a continuous time $t \in [0,1]$ with $\boldsymbol{x}(t) \in \mathbb{R}^n$, where n denotes the dimension of the image. We sample $\boldsymbol{x}(0)$ from unknown data distribution $p_0(\mathbf{x})$ and perturb the data points with a stochastic process over time $[0,1]$ such that $\boldsymbol{x}(1) \sim p_1(\mathbf{x})$, with $p_1(\mathbf{x})$ is close to a predefined noise distribution . This process is governed by an Itô stochastic differential equation (SDE) (Song et al., 2020) given by

$$d\boldsymbol{x}_t = \hat{f}(t)\boldsymbol{x}_t dt + \hat{g}(t)d\mathbf{w}_t \tag{1}$$

where $\hat{f} : \mathbb{R}^n \to \mathbb{R}^n$ denotes the drift coefficient, $\hat{g}(t) : \mathbb{R} \to \mathbb{R}$ defines a diffusion coefficient, and $\mathbf{w}_t \in \mathbb{R}^n$ denotes a Wiener process. The perturbation process SDE in Equation (1)

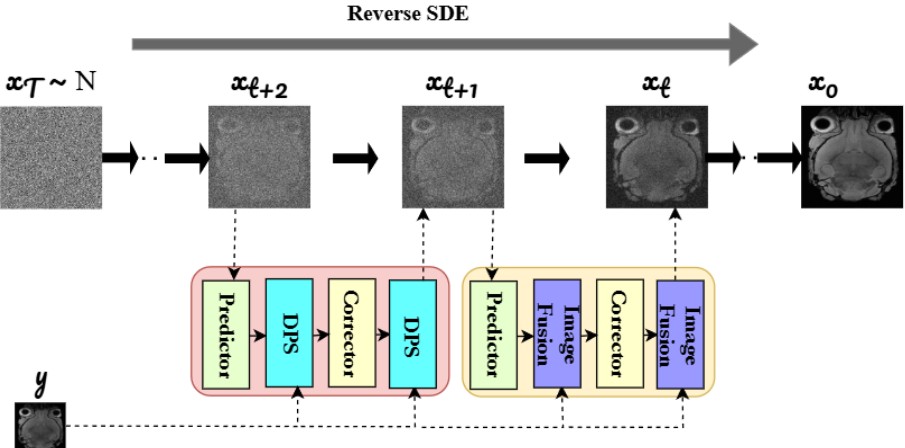

Figure 1: Overview of the proposed score-based diffusion method for reconstructing HR MRI images. Starting from pure noise $x_T$, we obtain $x_0$ through alternating reverse SDE numerical solver and image fusion step.

can be associated with the following reverse SDE given by Anderson's theorem (Anderson, 1982)

$$dx_t = \left[ \hat{f}(t)x_t - \hat{g}(t)^2 \nabla_{x_t} \log p_t(x_t) \right] dt + \hat{g}(t)d\overline{w}_t \tag{2}$$

where $\overline{w}_t$ is a Wiener process running backward in time from 1 to 0, and $dt$ is an infinitesimal negative timestep. To solve Equation (2), we require the score function of $p_t(x_t)$, i.e. $\nabla_{x_t} \log p_t(x_t)$, which can be estimated by the time-conditioned neural network $s_\theta(x_t, t)$ .

To solve Equation (2), we use numerical solvers such as Euler-Maruyama discretization, and Predictor-Corrector (PC) solvers (Song et al., 2020). With a score-based diffusion model, we can generate unconditional samples from the prior distribution $p_0(x)$ of HR images $x$. However, to obtain HR images from LR images, we need to sample from the posterior distribution $p_0(x|y)$ where $y$ denotes LR image.

## 2.2. Diffusion Posterior Sampling (DPS)

For image super-resolution, we aim to recover the unknown high-resolution image $x \in \mathbb{R}^n$ from a degraded measurement $y \in \mathbb{R}^m$ , which is modeled as:

$$y = Hx_0 \tag{3}$$

$H \in \mathbb{R}^{m \times n}$ is an unknown degradation process. We can formulate the reverse SDE to sample from the posterior distribution by modifying the SDE in Equation (2) as follows:

$$dx_t = \left[ \hat{f}(t)x_t - \hat{g}(t)^2 \left( \nabla_{x_t} \log p_t(x_t) + \nabla_{x_t} \log p_t(y|x_t) \right) \right] dt + \hat{g}(t)d\overline{w}_t \tag{4}$$

In Equation (4), we need to compute two terms: the score function $\nabla_{\mathbf{x}_t} \log p_t(\mathbf{x}_t)$ and the likelihood $\nabla_{\mathbf{x}_t} \log p_t(\boldsymbol{y}|\boldsymbol{x}_t)$. We can compute the first term using the pre-trained score function $s_\theta$. The second term can be obtained through Diffusion Posterior Sampling (DPS) (Chung et al., 2023) that provides an approximation of the likelihood which does not have an analytical formulation.

The posterior mean in the case of VE-SDE (Song et al., 2020) for $p(\boldsymbol{x}_0|\boldsymbol{x}_t)$ can be obtained through the Tweedie's approach (Efron, 2011; Chung et al., 2023) such that the posterior mean becomes as $\hat{\boldsymbol{x}}_0 \simeq \boldsymbol{x}_t + b_t^2 \mathbf{s}_{\boldsymbol{\theta}}(\mathbf{x}_t, t)$.

With this posterior mean, we can approximate the gradient of the log-likelihood

$$\nabla_{\boldsymbol{x}_t} \log p\left(\boldsymbol{y}|\boldsymbol{x}_t\right) \simeq \nabla_{\boldsymbol{x}_t} \log p\left(\boldsymbol{y}|\hat{\boldsymbol{x}}_0\right) \tag{5}$$

## 3. Method

### 3.1. DPS for MRI super-resolution

The forward model in Equation (3) can be alternatively formulated as:

$$\boldsymbol{y} \sim \mathcal{N}\left(\boldsymbol{y}|\boldsymbol{H}^f \boldsymbol{x_0}, \boldsymbol{I}\right) \tag{6}$$

where $\boldsymbol{H}^f \in \mathbb{R}^{m \times n}$ is an unknown downsampling block, and the forward model is assumed to follow a Gaussian distribution. Then, the likelihood function takes the form

$$p\left(\boldsymbol{y}|\boldsymbol{x}_0\right) = \frac{1}{\sqrt{(2\pi)^m}} \exp\left[-\frac{\|\boldsymbol{y} - \boldsymbol{H}\boldsymbol{x}_0\|_2^2}{2}\right] \tag{7}$$

Differentiating Equation (7) with respect to $\mathbf{x}_t$, using Equation (5), we get

$$\nabla_{\boldsymbol{x}_t} \log p\left(\boldsymbol{y}|\boldsymbol{x}_t\right) \simeq -\nabla_{\boldsymbol{x}_t} \|\boldsymbol{y} - \boldsymbol{H}\left(\hat{\boldsymbol{x}}_0\left(\boldsymbol{x}_t\right)\right)\|_2^2 \tag{8}$$

where we write $\hat{\boldsymbol{x}}_0 := \hat{\boldsymbol{x}}_0\left(\boldsymbol{x}_t\right)$, such that $\hat{\boldsymbol{x}}_0$ is a function of $\boldsymbol{x}_t$. Consequently, calculating the gradient $\nabla_{\boldsymbol{x}_t}$ is equivalent to performing backpropagation through the neural network. Finally, to calculate the gradient of marginal distribution $\nabla_{\boldsymbol{x}_t} \log p_t\left(\boldsymbol{x}_t|\boldsymbol{y}\right)$, we sum up the gradient of log-likelihood $\nabla_{\boldsymbol{x}_t} \log p\left(\boldsymbol{y}|\boldsymbol{x}_t\right)$ and use $\mathbf{s}_{\boldsymbol{\theta}}(\mathbf{x}_t, t)$ for prior to obtain

$$\nabla_{\boldsymbol{x}_t} \log p_t\left(\boldsymbol{x}_t|\boldsymbol{y}\right) \simeq \boldsymbol{s}_\theta\left(\boldsymbol{x}_t, t\right) - \rho \nabla_{\boldsymbol{x}_t} \left\|\boldsymbol{y} - \hat{\boldsymbol{H}}\left(\hat{\boldsymbol{x}}_0\right)\right\|_2^2 \tag{9}$$

where $\rho$ is the step size and a chosen downsampling block, $\hat{\boldsymbol{H}}$ (e.g., Bicubic), is used to estimate a downsampled $\hat{\boldsymbol{x}}_0$. Notably, the choice of downsampling block for data consistency does not affect the evaluation metrics, as demonstrated in B. This differs from supervised methods that are trained on a simulated dataset, where the models become biased toward the specific downsampling kernel used (Shrestha et al., 2023). Using only DPS for data consistency requires numerous computationally expensive backpropagations, resulting in a prolonged image reconstruction time.

## 3.2. Iterative Image Fusion during Sampling

To overcome the high computational cost of backpropagation, we propose a hybrid sampling approach that alternates between backpropagation and a more efficient image fusion strategy. Specifically, at time $t$, we employ DPS to estimate $\mathbf{x}_{t-1}$ and $\hat{\mathbf{x}}_0$. For the subsequent step, we utilize an image fusion technique that integrates $\mathbf{x}_{t-1}$, with $\mathbf{y}$. We decompose an image into multiple subbands at different scales, including low-low, low-high, high-low, and high-high bands. An effective integration strategy is to select wavelet coefficients from both the LR and HR images: the low-low band from the LR image and the other coefficients from the HR image. Then, the Inverse Discrete Wavelet Transform (IDWT) is applied to the combined coefficients to construct a fused image. This fusion approach effectively harmonizes the high-resolution features of $\mathbf{x}_{t-1}$ with the measurement data $\mathbf{y}$, yielding $\mathbf{x}_{t-2}$ that ensures data consistency.

We employ a wavelet-based fusion technique at time $t-1$ to integrate the high-resolution features of $\mathbf{x}_{t-1}$ with the upscaled low-resolution image $\mathbf{y}$ which can modeled as

$$W\left(\mathbf{x}_{t-1}\right) = \left\{A_x, D_x\right\}, W(\mathbf{y}) = \left\{A_y, D_y\right\} \tag{10}$$

where $W$ denotes the wavelet decomposition operation, $A$ represents the approximation (low-frequency) coefficients, and $D$ represents the detail (high-frequency) coefficients. The fusion of these components is defined as $A_{\text{fused}} = A_y, \quad D_{fused} = D_x$.

Finally, the fused image $\mathbf{x}_{t-2}$ is reconstructed from the fused wavelet coefficients using the inverse wavelet transform: $\mathbf{x}_{t-2} = W^{-1}\left(A_{\text{fused}}, D_{fused}\right)$.

This method offers a computationally efficient solution, as it bypasses the need for backpropagation. As outlined in Algorithm 1, our method combines the predictor and corrector steps of the PC algorithm with $k$ skip steps for image fusion.

---

**Algorithm 1** SRMRI Predictor-Corrector (PC) Sampling

---

**Require:** $s_\theta$, $N$, $M$, $\boldsymbol{y}$, $k$ $\triangleright k$ : skip steps
1: $\boldsymbol{x}_N \sim \mathcal{N}\left(\mathbf{0}, \sigma_T^2 \boldsymbol{I}\right)$
2: **for** $i = N$ **to** $1$ **do**
3:    $\boldsymbol{x}'_{i-1} \leftarrow \text{Predictor}\left(\boldsymbol{x}_i, \sigma_i, \sigma_{i-1}\right)$
4:    **if** $x \mod k = 0$ **then**
5:       $\hat{\boldsymbol{x}}_0 \leftarrow \boldsymbol{x}_i + \sigma_i^2 \mathbf{s}_{\boldsymbol{\theta}}(\mathbf{x}_i, \sigma_i)$
6:       $\boldsymbol{x}_{i-1} \leftarrow \boldsymbol{x}'_{i-1} - \rho\nabla_{\boldsymbol{x}_i} \left\|\boldsymbol{y} - \hat{H}(\hat{\boldsymbol{x}}_0)\right\|_2^2$
7:    **else**
8:       $Ax_{i-1}, Dx_{i-1} \leftarrow W\left(\boldsymbol{x}'_{i-1}\right); Ay, Dy \leftarrow W\left(\boldsymbol{y}\right)$
9:       $\boldsymbol{x}_{i-1} = W^{-1}\left(Ay, Dx_{i-1}\right)$
10:    **end if**
11:    **for** $j = 1$ **to** $M$ **do**
12:       $\boldsymbol{x}_{i-1} \leftarrow \text{Corrector}\left(\boldsymbol{x}_{i-1}, \sigma_{i-1}\right)$
13:    **end for**
14: **end for**
15: **return** $\boldsymbol{x}_0$

---

## 4. Experiments

We assess the effectiveness of our algorithm outlined by Algorithm 1 and compared it with other unsupervised and supervised learning-based baselines. Additional details about implementation can be found in A.1.

### 4.1. Dataset

We acquired brain images from 10 Alzheimer's disease (AD) mice with a 5xFAD background using a 9.4 T magnet with a 30-cm bore (Bruker-BioSpec 94/30, Billerica, MA). A 3D gradient echo (GRE) pulse sequence was performed at both 25 $\mu$m and 50 $\mu$m isotropic resolution. The field of view (FOV) was set to 18.0 mm $\times$ 12.8 mm $\times$ 7.6mm, flip angle of 45°, bandwidth (BW) of 125 kHz, and repetition time (TR) of 100 ms. We used the 25 $\mu$m resolution volumes, measuring 720 $\times$ 512 $\times$ 304, as high-resolution (HR) and the 50 $\mu$m resolution volumes, measuring 360 $\times$ 256 $\times$ 152, as low-resolution (LR). For training, We acquired image slices along the coronal plane, resulting in HR images with dimensions of 720$\times$512. We removed the first and last fifteen slices from each volume to exclude noise-only data and improve training. This results in approximately 3k slices of training data. For evaluation, we used the HR and LR volumes from 4 subjects. The 25 $\mu$m scans had the voxel spacing of (0.025, 0.025, 0.025), while the 50 $\mu$m scans had a spacing of (0.05, 0.05, 0.05). It is important to note that our training and evaluation dataset consists of LR-HR pairs obtained directly from the scanner. Due to hardware and physical constraints, acquiring perfectly matched pairs of HR and LR images is generally not feasible. Therefore, we formed test pairs by selecting a 50 $\mu$m image and identifying the closest 25 $\mu$m image from the same subject using a voting scheme based on three metrics: LPIPS (Zhang et al., 2018a), PCA (Jolliffe and Cadima, 2016), and SSIM (Wang et al., 2004). The pair with the most votes was chosen. Further details about datasets are provided in A.2.

### 4.2. Comparison study

#### 4.2.1. Unsupervised methods

We compared our approach with four unsupervised super-resolution methods baselines: DPS (Chung et al., 2023), manifold constrained gradients (MCG) (Chung et al., 2022b), KernelGAN (Bell-Kligler et al., 2020) + ZSSR (Shocher et al., 2017) and Score-SDE (Song et al., 2020). DPS, MCG, and Score-SDE are diffusion-based models that can be used for solving inverse problems. Comparisons were made using PSNR, SSIM, LPIPS, and inference time. One should note that standard metrics such as PSNR and SSIM may not fully capture perceptual quality in our dataset, given that LR and HR images do not align on a strict pixel-by-pixel basis. Although we report these metrics to provide a baseline comparison, their values should be interpreted with caution. In addition, we used more perceptually oriented metrics (LPIPS) to better reflect reconstruction quality.

Table 1 shows the results for unsupervised training, where our method, SMRI, outperforms other techniques across evaluation metrics. Additionally, it is worth noting that our method offers a faster inference with higher performance compared to existing unsupervised approaches that use diffusion models for solving inverse problems. In the case of Kernel-GAN + ZSSR, KernelGAN estimates the unknown downsampling kernel from the input LR

Table 1: Quantitative evaluation (PSNR, SSIM, LPIPS) of MRI super-resolution (Unsupervised) on scanner images. **Bold**: Best, under: second best. k : skip step.

| Method | PSNR ↑ | SSIM ↑ | LPIPS ↓ | time (s) ↓ |
|---|---|---|---|---|
| Score-SDE (Song et al., 2020) | 23.58 | 0.49 | 0.17 | 740 |
| DPS(Chung et al., 2023) | 24.23 | 0.58 | 0.11 | 1194 |
| MCG (Chung et al., 2022b) | 24.10 | 0.53 | 0.12 | 1198 |
| KernelGAN (Bell-Kligler et al., 2020) + ZSSR (Shocher et al., 2017) | 18.44 | 0.36 | 0.43 | **175** |
| **SRMRI (Ours)(k=2)** | **24.75** | **0.61** | **0.1** | 873 |
| **SRMRI (Ours)(k=3)** | 24.54 | 0.60 | 0.11 | 790 |

images, while ZSSR uses this learned kernel to perform zero-shot super-resolution leading to lower overall performance.

### 4.2.2. SUPERVISED METHODS

We also evaluated our method against the supervised techniques listed in Table 2, using the training dataset described in Section 4.1.

Table 2: Quantitative evaluation (PSNR, SSIM, LPIPS) of MRI super-resolution with sensor images (Supervised).

| Method | PSNR ↑ | SSIM ↑ | LPIPS ↓ |
|---|---|---|---|
| SRCNN (Dong et al., 2015) | 22.49 | 0.64 | 0.55 |
| DDBPN (Haris et al., 2018) | 25.16 | 0.64 | 0.38 |
| CARN (Ahn et al., 2018) | **26.31** | **0.65** | 0.35 |
| Swinir (Liang et al., 2021) | 25.88 | 0.62 | 0.36 |
| RCAN (Zhang et al., 2018b) | 25.18 | **0.65** | 0.31 |
| ESRGAN (Wang et al., 2018) | 26.11 | **0.65** | 0.34 |
| **SRMRI (Ours) (k=2)** | 24.75 | 0.61 | **0.1** |
| **SRMRI (Ours) (k=3)** | 24.54 | 0.60 | 0.11 |

Table 2 compares our method with supervised methods. Unlike unsupervised approaches, supervised methods require training pairs, which are often simulated. For these results, the supervised models were trained using LR-HR pairs from the scanner. Although the supervised methods achieve higher PSNR and SSIM on our dataset, they exhibit more artifacts and blurring in the reconstructed images (see Figure 2). We attribute this to their reliance on learning LR-HR correspondence during training to reduce specific loss functions (eg. pixel-based loss, adversarial loss). However, in our case, the degradation process is non-deterministic, and there is no exact pixel-to-pixel correspondence between the LR and HR pairs. As a result, these approaches produce lower perceptual quality despite better performance on standard numerical metrics. Further, supervised methods has to be retrained if the downsampling factor changes. However, our method can be applied to any down-

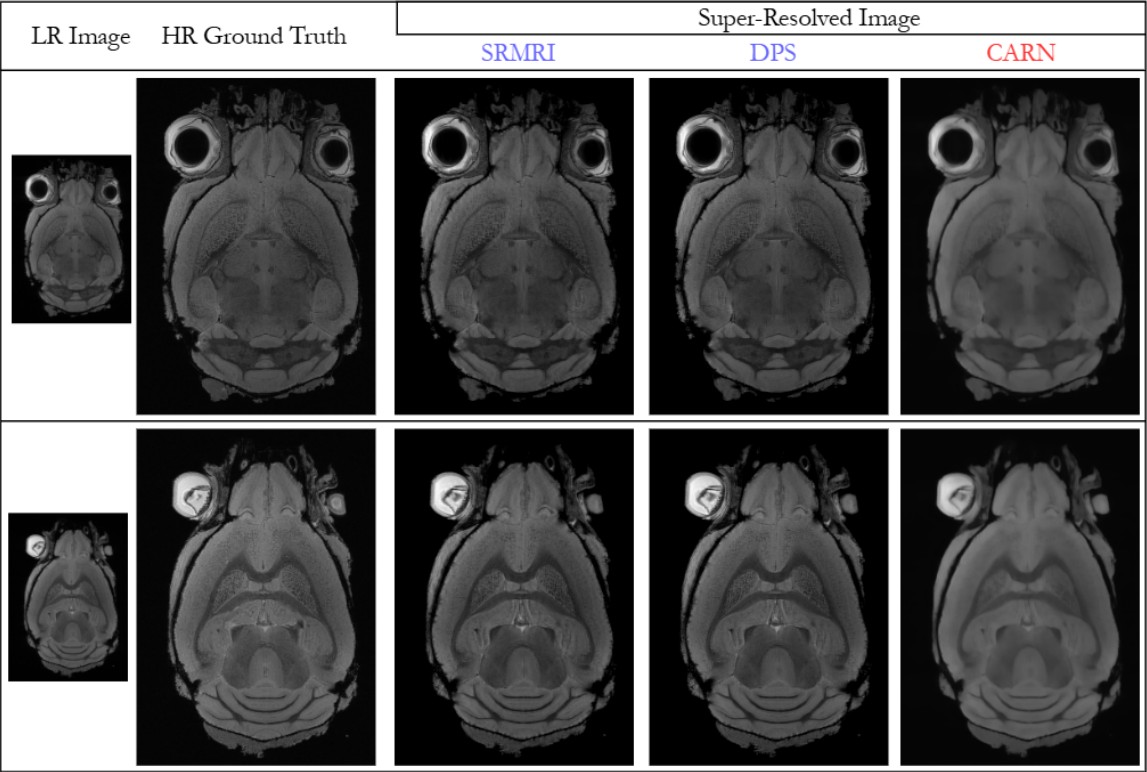

Figure 2: Examples of super-resolution (factor x2) results with scanner LR. You may zoom in to view more details. Blue: Unsupervised, Red: Supervised.

sampling factor. Figure 2 shows an example of super-resolutions (factor x2) from different supervised and unsupervised methods. More examples of super-resolutions can be seen at D. Finally, in Appendix C, we present a parameter-sensitivity study of the skip steps (k) used in our fusion strategy.

## 5. Discussion and Conclusion

In this paper, we introduce a blind MRI super-resolution method that alternates between image fusion and DPS during the diffusion-based sampling process. This approach reduces the computational load by replacing the computationally intensive DPS step with low-complexity image fusion technique. Our experimental results show that our method outperforms existing unsupervised approaches while offering a speed advantage. In comparison to supervised methods, which achieve higher metric values but introduce more artifacts and blurring, our approach produces better-quality images without the need for LR-HR training pairs, making it more practical for real-world scenario. Furthermore, our method can effectively recover complex and unknown degradations that may occur in real-world scenarios, even when the degradation is unknown or differs from the training data. Although we

down-sample intermediate images to maintain data consistency during sampling using the chosen degradation process, our method can be extended to construct multiple diffusion processes to learn priors for each component. This allows for posterior sampling even when the degradation operator is unknown (Chung et al., 2022a). Additionally, we introduce a dataset comprising both LR and HR MRI scans obtained directly from the scanner. To establish correspondence between the LR and HR pairs, we propose a voting scheme based on image quality metrics such as PSNR, SSIM, and LPIPS. This dataset can reduce dependence on simulated degradations for supervised super-resolution training. To the best of our knowledge, we are the first to evaluate model performance on such LR-HR pairs directly obtained from the scanner, providing a more accurate representation of performance in real-world scenarios.

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

## Appendix A. Experimental Details

### A.1. Implementation Details

#### A.1.1. Training of the score function

We used the implementation of the time-dependent score function model *ncsnpp* [1] (Song et al., 2020) as a score model. The model architecture consists of a time-conditioned U-Net, and the sub-block within U-Net is adopted from residual blocks of BigGAN (Brock et al., 2019). The network is conditioned on time (t) by incorporating Fourier features. These time-related features are combined with the original input features before being processed by the encoder.

The model was trained using a batch size of 4 and the Adam optimizer with standard hyperparameters ($\beta_1 = 0.9$ and $\beta_2 = 0.999$). To stabilize training, a linear learning rate warm-up was employed for the first 5000 steps, reaching a final learning rate of $2 \times 10^{-4}$. Gradient clipping was applied to prevent exploding gradients, and exponential moving averages were calculated for the model parameters. All experiments were conducted using PyTorch. The model was trained on the full training dataset for 1000 epochs, utilizing five RTX 3090 GPUs. This training process takes approximately five days of wall clock time.

#### A.1.2. Sampling

We modify the Predictor-Corrector (PC) sampler, as described in (Song et al., 2020), due to its superior performance in solving VE-SDE. The PC sampler consists of two components: the predictor, which is a numerical solver for the reverse-time SDE, and the corrector, where we use Langevin dynamics for the Markov chain Monte Carlo (MCMC) method. For the PC sampler, we used 2000 noise scales and 1 step of Langevin dynamics per noise scale. All the sampling steps outlined were executed on a single RTX 3090 GPU. In our experiments, we used the level two decomposition in the DWT for two levels of wavelet coefficients. Empirically, we found that bior4.4 performs best as the mother wavelet.

#### A.1.3. Code Availability

We will publish our code and dataset used in our experiments upon publication to boost reproducibility.

---

1. https://github.com/yang-song/score_sde_pytorch

## A.2. Dataset details

To prepare the dataset for training and evaluation of the supervised method, we used HR and LR image slices from two subjects, with isotropic resolutions of 25 $\mu$m and 50 $\mu$m, respectively. Although the HR and LR image volumes were acquired sequentially from the same subjects, no direct correspondence exists between the LR and HR slices, limiting their use as training pairs for supervised learning. Therefore, we employed three methods to pair the LR and HR images, forming a voting scheme where the image pair with the highest number of votes was selected. In cases where no consensus was reached during voting, the images from each method were visually inspected, and the closest pair was chosen. The following describes the implementation of those three methods.

### PCA based method

We implemented a Principal Component Analysis (PCA) based method to match LR images with their HR counterparts. Given the difference in resolution between the LR images (360x256) and the HR images (720x512), we first downsampled the HR images, reducing them to the same size as the LR images using cubic interpolation. Following this, PCA was employed to reduce the dimensionality of both the downsampled HR images and the original LR images. For each HR image, the Euclidean distance between its PCA-transformed representation and that of the LR image was computed. The HR image with the smallest distance was identified as the closest match to the LR image.

### LPIPS based method

In addition to the PCA-based approach, we implemented a method using Learned Perceptual Image Patch Similarity (LPIPS) to match LR images with their corresponding HR counterparts. LPIPS is a deep learning-based metric that evaluates perceptual similarity between images by comparing feature maps extracted from a pre-trained convolutional neural network (CNN). The LR and HR images were first resized to have identical dimensions. The perceptual distance between the LR and HR images was then calculated in the feature space, with lower LPIPS scores indicating higher perceptual similarity. The HR image with the lowest LPIPS distance was selected as the best match for the LR image.

### SSIM based Method

For our third method, we utilized the SSIM to assess the structural similarity between the LR and HR images. As with the other methods, the HR images were resized to match the dimensions of the LR images. The SSIM value was then calculated for each HR-LR image pair, with higher SSIM values indicating greater structural similarity. The HR image with the highest SSIM score was considered the closest match to the LR image.

## Appendix B. Impact of Downsampling Methods

In this section, we investigate the effect of using different downsampling kernels in our data consistency block with k = 2. Specifically, we tested the following methods: bicubic, linear, Lanczos2 (Wolberg, 1992 - 1990), Lanczos3 (Wolberg, 1992 - 1990), and box. Table B summarizes the results in terms of PSNR, SSIM, and LPIPS. We observe that the choice

of downsampling has a negligible impact on these evaluation metrics. Hence, any of these kernels can be employed without adversely affecting the final performance.

Table 3: Comparison of different downsampling methods.

| Method | PSNR ↑ | SSIM ↑ | LPIPS ↓ |
|---|---|---|---|
| Bicubic | 24.75 | 0.61 | 0.1 |
| Bilinear | 24.61 | 0.60 | 0.09 |
| Lanczos2 (Wolberg, 1992 - 1990) | 24.61 | 0.6 | 0.1 |
| Lanczos3 (Wolberg, 1992 - 1990) | 24.65 | 0.6 | 0.1 |
| Box | 24.72 | 0.61 | 0.1 |

As shown in Table B, the variation across downsampling methods is minimal, with differences typically within the margin of error. Consequently, for our main experiments, we chose the bicubic kernel as the default downsampling method for data consistency.

## Appendix C. Choice of skip steps (k)

We varied k from 1 (pure DPS) to 5 and observed that smaller values (2–3) provide the best trade-off between computational cost and reconstruction fidelity, as shown in Table C. Notably, k=2 yields higher PSNR and SSIM than k=1 (full DPS) while reducing inference time from 1194s to 873s. As k increases beyond 3, speed gains become marginal, and quality metrics slightly decrease. To further validate our image-fusion step, we removed fusion for k=2, thereby omitting data consistency during the alternating steps. As expected, performance dropped notably—both quantitatively (PSNR fell from 24.75 to 17.88) and qualitatively, where reconstructions diverged from the ground-truth HR images.

Table 4: Quantitative evaluation for different value of k

| k | PSNR ↑ | SSIM ↑ | LPIPS ↓ | time (s) ↓ |
|---|---|---|---|---|
| 1 (DPS) | 24.23 | 0.58 | 0.11 | 1194 |
| k=2 | 24.75 | 0.61 | 0.1 | 873 |
| k=3 | 24.54 | 0.60 | 0.11 | 790 |
| k=4 | 24.44 | 0.59 | 0.11 | 756 |
| k=5 | 24.19 | 0.54 | 0.11 | 729 |
| k=2 (without Image Fusion) | 17.88 | 0.36 | 0.39 | 727 |

## Appendix D. Additional Examples

### D.1. Unsupervised Methods

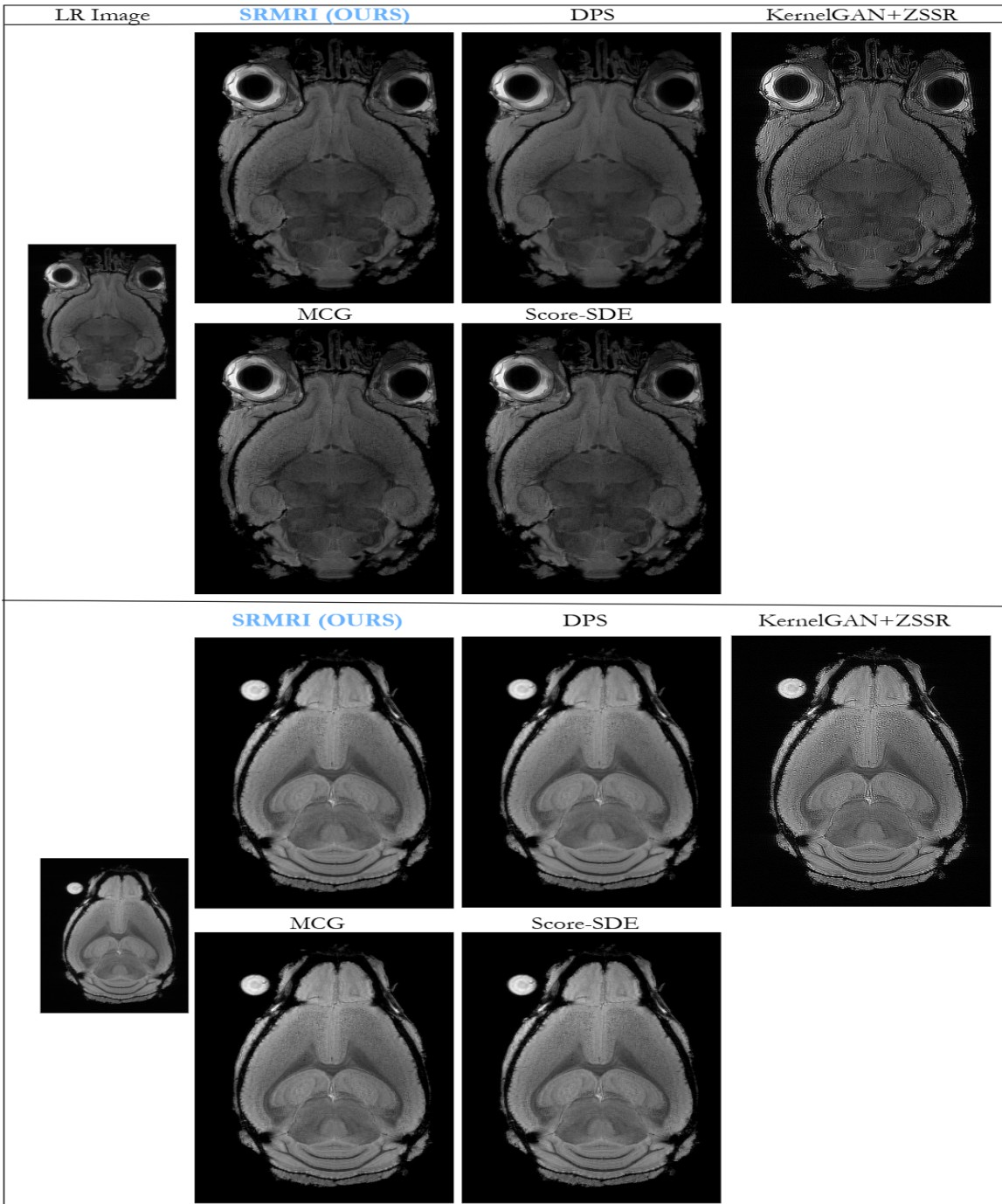

Figure 3: Examples of super-resolution (x2) with unsupervised methods.

## D.2. Supervised Methods

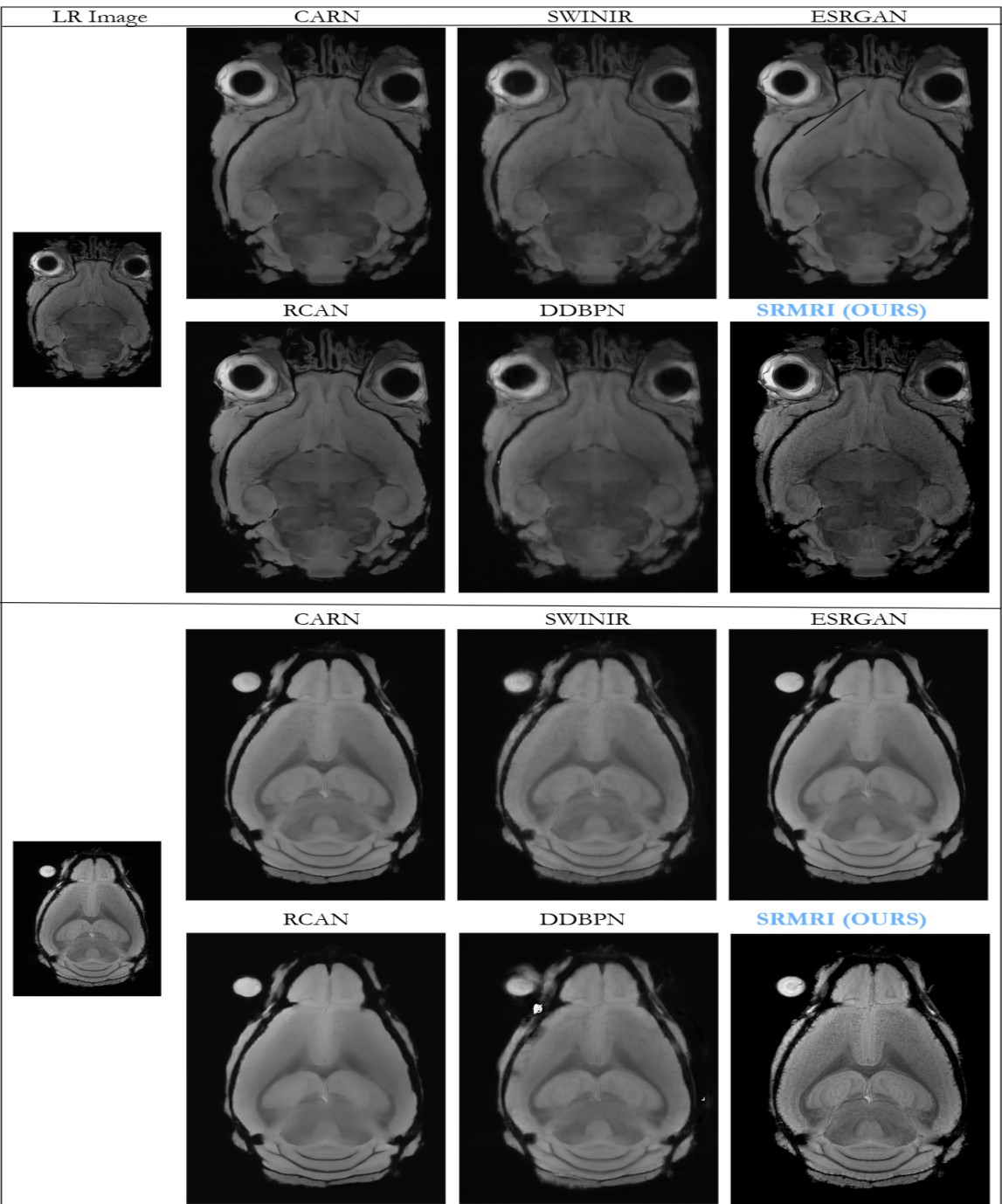

Figure 4: Examples of super-resolution (x2) with supervised methods. Blue: ours.

**HR**

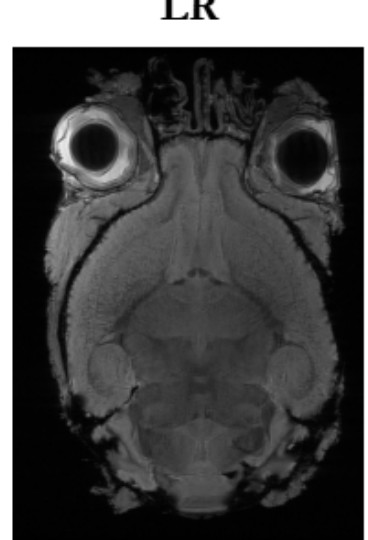

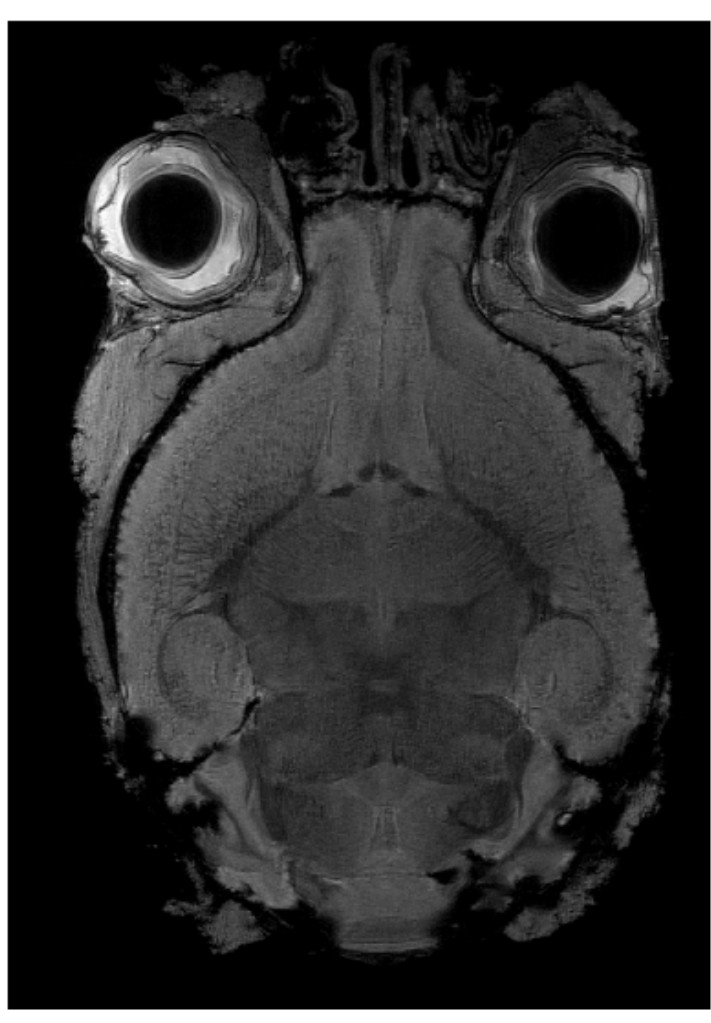

Figure 5: Full-resolution LR and super-resolved HR (scaled to half to fit).

