# OpenReview forum: "SRMRI: A Diffusion-Based Super-resolution Framework and Open Dataset for Blind MRI Super-Resolution"
_MIDL.io/2025/Conference — MIDL 2025 Poster_

### Official Review · Reviewer_baDf · 2025-02-16

**Confidence:** 4
**Preliminary Rating:** 4
**Recommendation:** Poster
**Final Rating:** 4

**Summary:**

This paper proposes a novel unsupervised MRI super-resolution framework that leverages a score-based diffusion model in combination with Diffusion Posterior Sampling (DPS) and an innovative iterative image fusion strategy to recover high-resolution (HR) images from low-resolution (LR) inputs without requiring paired training data. The method first learns the HR image prior via denoising score matching on HR scans and then alternates between backpropagation-based DPS steps and a computationally efficient wavelet-based image fusion technique, maintaining the benefits of DPS at a much lower inference cost. The authors also introduce a benchmark dataset consisting of scanner-obtained LR–HR pairs from mice. This offers a more realistic evaluation scenario compared to the common benchmarking on simulated degradations. Quantitative experiments show that the method outperforms standard DPS and other unsupervised methods. The method underperforms in reconstruction metrics against supervised methods, although the unsupervised method displays much better perceptual similarity which the authors show qualitatively to be linked to artefacts produced by supervised methods.

**Strengths:**

• Novelty: The proposed modification of existing DPS-based super-resolution is a novel addition that is clearly motivated by the inefficiency of DPS.
• Dataset Contribution: The introduction of the SRMRI dataset, comprising directly acquired scanner images, provides a valuable resource that reflects practical challenges better than simulated pairs.
• Comprehensive Experimental Evaluation: The paper rigorously compares the proposed method with both unsupervised and supervised baselines using multiple metrics (PSNR, SSIM, LPIPS, inference time) and illustrates qualitative improvements in perceptual quality.

**Weaknesses:**

• Quantitative Metric Gap: Although competitive in the unsupervised domain, the method still underperforms supervised approaches in traditional metrics (PSNR and SSIM).
• Limited Ablation on Fusion Strategy: While the image fusion step is a key component, the paper could benefit from more extensive ablation studies to quantify its specific contribution relative to full DPS iterations. Ablation of fusion strategy with other fusion methods could be explored.
• Dataset Size and Diversity: The SRMRI dataset is based on a limited number of subjects (10 for training and 4 for evaluation), which raises questions about the method’s generalisability across different MRI scanners, protocols, or patient populations. Evaluating on other datasets would provide more confidence in the general applicability of the proposed method. This could even be done to further highlight the shortcomings of supervised methods, e.g. showing training with one form of downsampling and testing on multiple different forms.
• Clarity on Pairing Strategy: The voting scheme used to form LR–HR pairs for evaluation is briefly described; additional details on its robustness and potential misalignment issues would strengthen the experimental analysis.

**Detailed Comments:**

• Image Fusion Details: A more in-depth analysis of the fusion strategy (e.g., sensitivity to the skip parameter k ) would clarify how it balances data consistency and computational efficiency.
• Pairing Methodology: Expanding the discussion on the voting scheme for LR–HR pairing - including potential edge cases or failure modes - would help assess the reliability of the evaluation.
• Generalization: Some discussion on how the method might extend to other MRI modalities or even non-MRI imaging domains would add value.
• Typo / correction: Table 2 shows (k=2) twice. I think second one should be (k=3)

**Justification Of The Final Rating:**

Thank you to the authors for the clarification regarding data preparation and parameter sensitivity. Concerns still remain regarding the dataset diversity/size, which was the justification cited for choosing a weak accept over, and so the original rating stands.

**Justification Of The Preliminary Rating:**

The paper presents a novel unsupervised MRI super-resolution framework that combines a score-based diffusion model with Diffusion Posterior Sampling (DPS) and a computationally efficient iterative image fusion strategy. This approach directly addresses the limitations of relying on simulated LR–HR pairs and the inefficiency of full DPS iterations. The introduction of the SRMRI dataset, featuring scanner-obtained LR–HR pairs from mice, provides a realistic benchmark that better reflects practical challenges in MRI acquisition. Quantitative experiments and qualitative analyses demonstrate that while the method underperforms in traditional metrics (e.g., PSNR and SSIM) compared to supervised methods, it achieves superior perceptual quality and competitive performance among unsupervised approaches. However, the limited data size and diversity of the evaluation limits the confidence in any conclusions drawn from the results. Overall, the paper’s innovative contributions and provision of a valuable dataset justify a weak accept.

**Questions To Address In The Rebuttal:**

• Parameter Sensitivity: How sensitive is the reconstruction quality to the choice of the skip step parameter  k  in the image fusion strategy?
• Fusion vs. Full DPS: Can the authors provide additional quantitative or qualitative evidence to isolate the impact of the image fusion steps compared to performing full DPS iterations?
• Dataset Robustness: Could the authors comment on how well the method generalizes across different scanners or MRI protocols, and whether they have any preliminary results on larger or more diverse datasets?
• Pairing Scheme Robustness: What is the error rate or variability in the LR–HR pairing obtained via the voting scheme, and how might this affect the evaluation of supervised methods?

**Special Issue:**

Yes

---

> ### Author Response · Authors · 2025-03-07
>
> We thank the reviewer for their thoughtful feedback. Below, we address individual concerns:
> ### Dataset Robustness
> We appreciate the concern about the dataset's diversity and size. Unfortunately, evaluating our method on a larger dataset was not feasible because paired low-resolution (LR) and high-resolution (HR) MRI scans are rare. Unlike natural images, MRI data with exact LR-HR pairs are not readily available in public repositories. Acquiring such data is challenging – it requires scanning the same subject twice to obtain two versions of the image at different resolutions. This is rarely done in practice, so there was no larger existing dataset we could use for additional experiments. Further, regarding different scanner or MRI protocols, if we have HR resolution images from such scanners or protocols, we can train/fine-tune our model on such to learn the data distribution and we can sample from the diverse distribution with consistency from LR image.
>
> ### Pairing Scheme Robustness
> Here we describe our pairing process in detail and discuss its limitations. We matched each LR slice with the corresponding HR slice through a semi-automated process. For a given LR image, we first used a script to find similarity measure (LPIPS/SSIM/PCA) to find the top three candidate HR images that could be its match. Then, from those top candidates, we selected the best matching HR image. In most cases, the correct match was from the automatic voting. When it wasn’t immediately clear, we performed a manual verification by visually checking alignment of anatomical structures from top three images from each method. We accept that this matching method is not perfect. Minor misalignments can occur because the LR and HR scans are acquired separately. Even with the same subject, there might be slight movement between scans. One obvious solution to improve alignment is to use image registration techniques to warp or align the LR image to the HR image. We carefully considered this but decided against it because most registration algorithms would upsample or resample the LR image as part of the alignment. In order to match the HR image's coordinate space, the LR image would be interpolated to a higher resolution. This interpolation would modify the LR image before it is fed into our model. Since our goal was to evaluate super-resolution on the raw scanner output, we worried that registration-based upsampling might affect the results. That said, we agree that improving the pairing further is a worthwhile direction for future work. One idea is to explore registration techniques that align anatomical structures without performing full image upscaling.
>
> ### Parameter Sensitivity
> We thank the reviewer for highlighting the parameter sensitivity of k in our image-fusion strategy. In our experiments (see Table 4 in the revised manuscript), we varied k from 1 (pure DPS) to 5 and observed that smaller values (2–3) provide the best trade-off between computational cost and reconstruction fidelity. Notably, k=2 yields higher PSNR and SSIM than k=1 (full DPS) while reducing inference time from 1194s to 873s. As k increases beyond 3, speed gains become marginal, and quality metrics slightly decrease. To further validate our image-fusion step, we removed fusion for k=2, thereby omitting data consistency during the alternating steps. As expected, performance dropped notably—both quantitatively (PSNR fell from 24.75 to 17.88) and qualitatively, where reconstructions diverged from the ground-truth HR images.

---

### Official Review · Reviewer_2U4p · 2025-02-18

**Confidence:** 5
**Preliminary Rating:** 2
**Final Rating:** 2

**Summary:**

This paper introduces a new blind MRI superresolution method.  It also provides a new open dataset.

**Strengths:**

The new dataset may be useful, although the fact that it is fully Fourier encoded is a limitation because it is hard to perform super-resolution with this kind of data.  Other forms of MRI superresolution are much more popular.

The method seems to be new, although it seems to be a simple assembly of existing techniques, and the novelty may be limited.

The paper uses a black-box performance evaluation, but there is limited insight or intuition about performance.  It is unclear when the new method is likely to be successful or when the new method is likely to fail.

**Weaknesses:**

The title of the manuscript focuses on the open-source dataset, but this is not the most significant the contribution of the paper.  The title does not mention the new unsupervised approach based on a score-based diffusion model, which seems to be the most significant contribution.

The paper seems to use an unrealistic model of resolution degradation in MRI, which seems to be driven by a poor understanding of MRI physics.  In MRI, resolution along the slice dimension is based on the slice excitation profile, while in-plane resolution is based on k-space coverage.  In 3D GRE, there is no slice-select dimension, and resolution will be very accurately modeled by convolution with a 3D sinc function point spread function.  The downsampling block in MRI is largely well-understood and very different from Gaussian blurring or bicubic downsampling, since the 3D sinc function will completely eliminate high-frequency information, unlike Gaussian blurring or bicubic downsampling which will simply attenuate the high frequency information.  It is unclear why the authors claim that the degradation model is unknown, particularly when using a sequence like 3D GRE with a large acquisition bandwdth, where field inhomogeneity is not an issue, and particularly when scanning ex vivo brains where patient motion will not occur.  The way that MRI is presented in this paper is misleading, and the problem that the authors are proposing to solve is not actually a problem in practice.

It is also not clear if the authors are aware that there are multiple forms of MRI superresolution problems.  The manuscript focuses on "blind" single-image super-resolution, while the earliest and most powerful MRI superresolution methods are based on acquiring multiple low-resolution images.  This is useful to explain in the literature review, since it will be useful to MRI novices and give expert readers more confidence that the authors are familiar with MRI.

Another major concern is that PSNR and SSIM and LPIPS can be insensitive to high-resolution image features, which is important when considering super-resolution problems.  It would be helpful to analyze performance using metrics that are more sensitive to resolution, for instance perturbation responses (DOI: 10.1002/mrm.28828)

The comparison methods used in this paper are mostly for generic super-resolution.  It is unclear why the authors have not compared against more methods designed specifically for MRI.

The SSIM values are all quite low (around 0.6), suggesting that the methods may not be practically useful.  Are these values accurate, or is the poor SSIM coming from a "hidden noise" problem?

**Detailed Comments:**

The mouse brains are not described accurately.  5xFAD mice have pathology that replicates certain aspects of Alzheimer's disease, but the mice do not actually have Alzheimer's disease.

Since the authors are focused on Fourier super-resolution (rather than slice super-resolution), they are encouraged to be aware of the large amount of existing literature on Fourier superresolution (DOI: 10.1016/B978-0-12-822726-8.00014-2) as well as the criticisms of Fourier superresolution (DOI: 10.1002/mrm.10203). Fourier superresolution is a much harder inverse problem than slice-based super-resolution.

**Justification Of The Final Rating:**

Unfortunately, it does not appear that the authors made revisions that would satisfy my original concerns, so my final rating remains the same.  The authors' rebuttal suggests that revisions were made, but changes are not marked, and the new manuscript still seems to suffer from all of the same concerns I raised previously.

MRI physics is really well understood, and it is really easy to accurately simulate low-resolution MRI data from high-resolution MRI data if you have an understanding of physics and also have access to k-space.  There are many people working on MRI super-resolution who do not understand MRI and do a bad job of mimicking MRI, but that does not mean that the problem is fundamentally difficult.  A lot of the issues the authors raise, like field inhomogeneity, improper acquisition parameters, patient motion, and insufficient time are not barriers to accurate simulation, particularly ex vivo with high-bandwidth acquisitions.  MRI physics is really well understood, and the description of the problem the authors are solving still suggests a poor understanding of MRI physics and MRI data.  I can appreciate that the authors acquired real data rather than using simulation, but that does not mean that they have described the problem accurately.

The authors have also not addressed novelty.  Improved performance is not particularly novel when the metrics are questionable and the comparisons are designed for different scenarios with very different point-spread function characteristics than truncated Fourier transform MRI.  Most super-resolution problems assume attenuation of high frequencies using a locally-compact kernel, while FT MRI has a  degradation kernel that is not compactly supported and eliminates high frequencies.  The methods also seem straightforward, not particularly novel.

No changes appear to have been made in relation to the comments about PSNR, SSIM, noise, resolution, or incorrectly describing 5xFAD mice as "Alzheimer's disease mice"

**Justification Of The Preliminary Rating:**

The submission has many major weaknesses.  There seem to be major misunderstandings about MRI and a lack of awareness of related literature.  The dataset is new, but may not be very useful, and the results are not very impressive.

**Questions To Address In The Rebuttal:**

Please address the issues listed under weaknesses and detailed comments.

---

> ### Author Response · Authors · 2025-03-07
>
> We thank the reviewer for their thoughtful feedback. Below we address individual concerns:
> ### Regarding the Title and Focus of the Paper:
> Based on reviewer’s comments and suggestions, we reconsidered the title of our paper to reflect both the methodological contributions and the dataset. After confirming with the Program Chair, we have revised the title to: “SRMRI: A Diffusion-Based Super-resolution Framework and Open Dataset for Blind MRI Super-Resolution”.
>
> ### Clarifying MRI Physics and Degradation Models
> We apologize for any confusion and would like to clarify the MRI acquisition physics underlying our approach. We would like to re-enforce that we acquired low-resolution and high-resolution MRI data at 50 µm and 25 µm  of same subjects using two separate scans by changing the bandwidth, matrix size while keeping the ROI same. No explicit down sampling filtering methods like Gaussian or bicubic was used to generate low-resolution and high-resolution image pairs for trainings. As the reviewer suggested, one of our thesis indeed is that deep learning models trained using synthetic down sampling filter could actually produce optimistic results but would be biased towards the explicit model used, but fail in real low-resolution images from the scanners, which we have reported in (DOI: 10.1109/BHI58575.2023.10313380).
>
> Most of the literature on MR super-resolution uses specific degradation methods to mimic the LR image. For instance, (DOI: 10.1109/TMI.2024.3483109) uses K-space zero padding (KSZP), (DOI: 10.1016/j.bspc.2023.104901, DOI: 10.1016/j.knosys.2022.108669) uses bicubic interpolation, (DOI: 10.1038/s41598-022-10298-6) select the central part of k-space and add Gaussian Noise to simulate LR images which provides optimistic result but fails in real low-resolution scanner images. We also want to re-iterate that the bicubic downsampling on generated image we have in the reverse diffusion process is only for data-consistency part to make the reconstructed/generated image consistent with our low-resolution image in hand. It was not used to generate low-resolution and high-resolution pair for training. Moreover, our data-consistency approach is agnostic to bicubic or Gaussian or other filtering and any other technique for dimension matching can be used to perform data consistency.
>
> We certainly agree with the reviewer’s comment that in 3D GRE sequences there are no inherent 2D slice selection gradients. In our case, de-correlate third dimension from 3D acquisitions by taking inverse-Fourier transform along first two dimensions to get slice by slice image. This was purely done only to tackle the computational cost in 3D Diffusion type model by converting it to slice by slice reconstruction.  This is not uncommon practice for illustration purposes in literatures.
> (doi:10.48550/arXiv.1805.03300)
>
> ### Choice of Evaluation Metrics:
> We totally agree with reviewer’s view on quantitative metrics. PSNR and SSIM are indeed not the best metric as both are somewhat related to pixel-by-pixel type metrics, while we do now have pixel by pixel LR and HR pairs. We selected PSNR, SSIM, only to reflect that our method does comparatively better than LR and other competing methods, but we agree with reviewers that the numbers may not reflect the actual metrics.   We have made this clear in our new version.
>
> ### Baseline Selection:
> In our evaluation, we aimed to include both state-of-the-art general super-resolution methods and known unsupervised approaches. For the supervised baselines, we chose popular deep learning SR architectures that have proven effective on natural images. Our reasoning was that if a medical-specific SR method is based on deep learning, a network like RCAN or SwinIR (properly trained on our MRI data) would likely be as good as specialized method. The reviewer is correct that we did not include MRI-specific SR algorithms from the literature because many of those methods are not easily available for reproduction on our dataset.
>
> ### Interpreting the SSIM Values:
> The SSIM values are lower in our study mainly because of the nature of our test data – these are real LR and HR image pairs that are not perfectly aligned or identical. We mention in the paper that acquiring perfectly registered LR-HR pairs in MRI is generally infeasible, so we used a best-match approach. This means that some discrepancy is always present.
>
> ### Mouse Brains:
> The 5xFAD mouse model is widely recognized as a transgenic model that replicates certain aspects of Alzheimer’s disease pathology. However, as the reviewer correctly pointed out, these mice do not fully resemble the complete spectrum of human Alzheimer’s disease. However, our primary goal is to develop and evaluate super-resolution methods for MRI; to enhance the resolution of lower-resolution scans to better reveal anatomical and pathological details
>
> We are grateful for the reviewer’s insightful comments.

---

### Official Review · Reviewer_s243 · 2025-02-22

**Confidence:** 3
**Preliminary Rating:** 4
**Recommendation:** Poster
**Final Rating:** 4

**Summary:**

This paper addresses the challenge of MRI super-resolution (SR) by proposing an unsupervised method that circumvents the need for paired low- and high-resolution (LR/HR) training data. At the core is a score-based diffusion model, which learns to generate high-resolution images by modeling the underlying data distribution of MRI scans. Once trained, the model employs an iterative Diffusion Posterior Sampling (DPS) procedure to refine the images, alternating between computationally expensive backpropagation steps and a more cost-effective wavelet-based image fusion strategy. This dual-step process helps reduce the heavy computational burden typically associated with diffusion-based reconstruction. The authors validate their approach on a private dataset of MRI brain scans from ten Alzheimer’s disease (AD) mice, thereby demonstrating feasibility in a controlled, real-world scenario.

**Strengths:**

1. The paper tackles the important problem of MRI super-resolution, using unpaired data to train the SR model.
2. The authors boost reproducibility by publicly making their code and creating a dataset available.
3. The method is systematically evaluated against supervised and unsupervised baselines, underscoring its relative advantages and practicality. The author also compared different sampling methods for the data consistency block in their model architecture with k = 2 in the Appendix.
4. The authors modified the existing MRI SR method by using an iterative Diffusion Posterior Sampling (DPS) procedure to refine the images, alternating between computationally expensive backpropagation steps and a more cost-effective wavelet-based image fusion strategy, which reduces the heavy computational burden.

**Weaknesses:**

1. Experiments focus on a private dataset from only ten Alzheimer’s disease (AD) mice, restricting the broader applicability and statistical power of the results.
2. By extracting 2D slices along the coronal plane, the approach does not leverage the full volumetric information inherent to 3D MRI, which may leave untapped potential for more robust reconstruction.

**Detailed Comments:**

I suggest increasing the resolution of Figure 1 and Figure 2 to enhance clarity and ensure precise visual inspection, as the current lower-resolution images appear slightly blurred.

**Justification Of The Final Rating:**

Within the limited time for rebuttal, I appreciate the authors’ efforts to address my concerns. Nevertheless, I have maintained my previous rating. However, I also note that Reviewer 2U4p has raised some important questions that warrant further attention.

**Justification Of The Preliminary Rating:**

I like the question this paper is trying to solve that training Super-Resolution can take unpaired samples. The author modified the existing method (Chung et al., 2023) by using an iterative Diffusion Posterior Sampling (DPS) procedure to refine the images, alternating between computationally expensive backpropagation steps and a more cost-effective wavelet-based image fusion strategy, which reduces the heavy computational burden. Extensive comparison methods have been conducted with supervised and unsupervised baselines.

**Questions To Address In The Rebuttal:**

I would recommend the author conduct the experiments on a larger public dataset.

**Special Issue:**

No

---

> ### Author Response · Authors · 2025-03-07
>
> We thank the reviewer for their thoughtful feedback. Below we address the concerns regarding:
> ### Experiments focus on a private dataset and conduct experiments on a larger public dataset
> We chose private mouse brain MRI at 9.4 T because it provided an opportunity to get high-resolution ground truth (25 µm isotropic is far beyond clinical resolution). While this is a different domain than human MRI, the fundamental SR challenge is the same (recovering lost spatial detail). To our knowledge, our attempt is one of the first to create a low-resolution and high-resolution image dataset both from MR scanners directly, unlike other datasets where low-resolution images are mimicked through predefined filtering techniques such as bicubic, Gaussian filtering, which do not correctly represent the actual low-resolution MR images from scanners, providing optimistic results on synthetic low-resolution.
> Experiments on a larger dataset is unfortunately not feasible in our case due to the scarcity of matched low-resolution (LR) and high-resolution (HR) image pairs obtained from the scanner directly. This process is difficult and time-consuming. As a result, large-scale collections of true LR-HR pairs do not exist for our problem. While one could simulate a larger dataset by artificially degrading high-resolution images to create synthetic LR inputs, doing so would directly contradict the main claim of our paper. Our work argues that training on data generated by specific degradation models (e.g. bicubic down sampling or blurring) introduces bias into super-resolution models (DOI: 10.1109/BHI58575.2023.10313380). We therefore chose to stick with scanner-acquired data to maintain the integrity of our approach. If the reviewer suggests including experiments with a simulated dataset would be useful, we can provide the results in an appendix section.
>
> ### Figure Quality and Presentation
>  Due to page limits and format requirements, we had to downsize the figures to fit the layout for the reduced resolution of Figure 1 and Figure 2 in the main body of the paper. To address this issue, we provide full resolution of LR and HR in the appendix (figure 5) of the revised paper.
>
> ### Limitations with 2D slices
> We appreciate the reviewer’s concern regarding our decision to perform super-resolution on individual 2D slices rather than the entire 3D volume. We followed this approach due to the high computational requirements of diffusion models for such high-resolution images (we would have to load a 720×512×304 tensor into the GPU and perform diffusion on that entire tensor, which was not feasible in our computational resources.  Our goal in this paper is to illustrate how Diffusion based model can be used for super-resolution in MR imaging without using pixel-to-pixel types of loss and without explicitly using the synthetic down-sample approach such as Gaussian filtering or bicubic filtering to generate LR and HR image pairs for training. However, we also want to re-enforce that given computational resources, our approach can be extended to full 3D super-resolution are possible by leveraging 3D patch-by-patch . Recent work (DOI 10.48550/arXiv.2211.10655) has demonstrated that 2D pre-trained models can also be applied to 3D medical imaging tasks, such as sparse-view or limited-angle tomography, as well as compressed sensing MRI.
> We appreciate the opportunity to clarify these points and are confident that our revisions will address the reviewer's comments while underscoring the strengths of our approach.

---

### Author Rebuttal · Authors · 2025-03-08

**Rebuttal:**

We would like to extend our sincere gratitude for the thoughtful and constructive feedback provided. We have revised our manuscript to address all comments and suggestions, including a change of the paper title to more accurately reflect our study’s scope.

We conducted further experiments on our parameter sensitivity (k) to illustrate its effect on both computational efficiency and reconstruction fidelity.

One question shared by multiple reviewers is regarding testing our method on large datasets. We acknowledge that the size and diversity of our current dataset could be improved. However, the availability of paired low-resolution (LR) and high-resolution (HR) MRI data is limited due to the practical challenges of scanning the same subject multiple times at different resolutions. This is rarely performed in clinical or research workflows, resulting in a scarcity of publicly available, large-scale paired MRI datasets.

We believe these revisions strengthen the manuscript, and we appreciate your time and expertise in reviewing our work.

**Supporting Material:**

/attachment/eafc7d772baf78948284f507f2a3f3416d4abfe1.pdf

---

### Meta-Review · Area_Chair_HoMW · 2025-03-17

**Recommendation:** Reject
**Confidence:** 5

**Metareview:**

The reviewers appreciate the new dataset, which will be a useful resource for testing MRI super-resolution (SR) algorithms in real-world scenarios – rather than synthetically downsampled images. However, and even after the rebuttal, there are serious concerns with the proposed SR method, specifically: working in 2D (even the authors acknowledge in their rebuttal that they could have trained on patches to overcome GPU memory limits), inadequate modeling of resolution, limited accuracy of results, and lack of novelty. I am recommending rejection but, if the program chairs want to rescue this paper for the dataset aspect, I would not oppose to it.